# Host-Pathogen Adhesion as the Basis of Innovative Diagnostics for Emerging Pathogens

**DOI:** 10.3390/diagnostics11071259

**Published:** 2021-07-14

**Authors:** Alex van Belkum, Carina Almeida, Benjamin Bardiaux, Sarah V. Barrass, Sarah J. Butcher, Tuğçe Çaykara, Sounak Chowdhury, Rucha Datar, Ian Eastwood, Adrian Goldman, Manisha Goyal, Lotta Happonen, Nadia Izadi-Pruneyre, Theis Jacobsen, Pirjo H. Johnson, Volkhard A. J. Kempf, Andreas Kiessling, Juan Leva Bueno, Anchal Malik, Johan Malmström, Ina Meuskens, Paul A. Milner, Michael Nilges, Nicole Pamme, Sally A. Peyman, Ligia R. Rodrigues, Pablo Rodriguez-Mateos, Maria G. Sande, Carla Joana Silva, Aleksandra Cecylia Stasiak, Thilo Stehle, Arno Thibau, Diana J. Vaca, Dirk Linke

**Affiliations:** 1BioMérieux, Open Innovation & Partnerships, 38390 La Balme Les Grottes, France; manisha.goyal@biomerieux.com; 2Biomode, 4715-330 Braga, Portugal; carina.almeida@iniav.pt; 3Institut Pasteur, Structural Biology and Chemistry, 75724 Paris, France; bardiaux@pasteur.fr (B.B.); nadia.izadi@pasteur.fr (N.I.-P.); theis.jacobsen@pasteur.fr (T.J.); michael.nilges@pasteur.fr (M.N.); 4Department of Biological Sciences, University of Helsinki, 00014 Helsinki, Finland; sarah.barrass@helsinki.fi (S.V.B.); sarah.butcher@helsinki.fi (S.J.B.); a.goldman@leeds.ac.uk (A.G.); 5Centre for Nanotechnology and Smart Materials, 4760-034 Vila Nova de Famalicão, Portugal; tcaykara@centi.pt (T.Ç.); csilva@centi.pt (C.J.S.); 6Division of Infection Medicine, Department of Clinical Sciences, Lund University, 22242 Lund, Sweden; sounak.chowdhury@med.lu.se (S.C.); lotta.happonen@med.lu.se (L.H.); johan.malmstrom@med.lu.se (J.M.); 7BioMérieux, Microbiology R&D, 38390 La Balme Les Grottes, France; rucha.datar@biomerieux.com; 8Eluceda, Burnley BB11 5UB, UK; Ian@Eluceda.com; 9School of Biomedical Sciences, University of Leeds, Leeds LS2 9JT, UK; P.H.Johnson@leeds.ac.uk (P.H.J.); bsark@leeds.ac.uk (A.K.); bsjlev@leeds.ac.uk (J.L.B.); bsamali@leeds.ac.uk (A.M.); p.a.millner@leeds.ac.uk (P.A.M.); S.Peyman@leeds.ac.uk (S.A.P.); 10Institute for Medical Microbiology and Infection Control, University Hospital, Goethe-University, 60596 Frankfurt am Main, Germany; Volkhard.Kempf@kgu.de (V.A.J.K.); arno.thibau@kgu.de (A.T.); Diana.Vaca@kgu.de (D.J.V.); 11Department of Biosciences, University of Oslo, 0316 Oslo, Norway; ina.meuskens@ibv.uio.no; 12School of Mathematics and Physical Sciences, University of Hull, Hull HU6 7RX, UK; N.Pamme@hull.ac.uk (N.P.); P.Rodriguez-Mateos@hull.ac.uk (P.R.-M.); 13CEB—Centre of Biological Engineering, University of Minho, 4710-057 Braga, Portugal; lrmr@deb.uminho.pt (L.R.R.); msande@centi.pt (M.G.S.); 14Interfaculty Institute of Biochemistry, University of Tübingen, 72076 Tübingen, Germany; aleksandra.stasiak@uni-tuebingen.de (A.C.S.); thilo.stehle@uni-tuebingen.de (T.S.)

**Keywords:** adhesin, receptor, infectious diseases, diagnostics

## Abstract

Infectious diseases are an existential health threat, potentiated by emerging and re-emerging viruses and increasing bacterial antibiotic resistance. Targeted treatment of infectious diseases requires precision diagnostics, especially in cases where broad-range therapeutics such as antibiotics fail. There is thus an increasing need for new approaches to develop sensitive and specific in vitro diagnostic (IVD) tests. Basic science and translational research are needed to identify key microbial molecules as diagnostic targets, to identify relevant host counterparts, and to use this knowledge in developing or improving IVD. In this regard, an overlooked feature is the capacity of pathogens to adhere specifically to host cells and tissues. The molecular entities relevant for pathogen–surface interaction are the so-called adhesins. Adhesins vary from protein compounds to (poly-)saccharides or lipid structures that interact with eukaryotic host cell matrix molecules and receptors. Such interactions co-define the specificity and sensitivity of a diagnostic test. Currently, adhesin-receptor binding is typically used in the pre-analytical phase of IVD tests, focusing on pathogen enrichment. Further exploration of adhesin–ligand interaction, supported by present high-throughput “omics” technologies, might stimulate a new generation of broadly applicable pathogen detection and characterization tools. This review describes recent results of novel structure-defining technologies allowing for detailed molecular analysis of adhesins, their receptors and complexes. Since the host ligands evolve slowly, the corresponding adhesin interaction is under selective pressure to maintain a constant receptor binding domain. IVD should exploit such conserved binding sites and, in particular, use the human ligand to enrich the pathogen. We provide an inventory of methods based on adhesion factors and pathogen attachment mechanisms, which can also be of relevance to currently emerging pathogens, including SARS-CoV-2, the causative agent of COVID-19.

## 1. Introduction

Infectious diseases, particularly those caused by potentially lethal viruses and antibiotic-resistant bacteria, were one of the key issues on the agenda of the G7-Summit in Germany in 2015 (https://www.g7uk.org/new-international-approach-to-combat-emerging-health-threats-as-crucial-g7-health-talks-begin/, accessed on 21 June 2021) (see Table 1 for a review of the currently most relevant antibiotic resistant microorganisms). After the summit, every major health authority, including the World Health Organization (WHO), confirmed that the (re-)emergence of infectious diseases in general and the decreasing efficacy of antimicrobials are major medical concerns, as antimicrobial therapies are starting to fail (Table 1) and deadly viruses cause serious global outbreaks [1]. Moreover, it is estimated that around 30% of bacteria responsible for hospital associated infections are antibiotic resistant, with the number of infections being about nine million each year in Europe alone [2]. Clinical misdiagnosis can lead to antibiotic resistance when proper diagnostic methods are not used and patients are prescribed unnecessary treatments [3]. Thus, novel diagnostic tests are urgently needed to prevent (re-)emerging infections and to better treat infections by clinically relevant antibiotic resistant pathogens. Therefore, continued academic and industrial investment in new in vitro diagnostic (IVD) tests is urgently required to tackle antimicrobial resistance (AMR), (multi)drug-resistance (MDR) or even pan-resistance (PDR) [4,5,6].

Clearly, prior to infection, pathogens colonize their host organisms via adhesion: the binding of microbial molecules—adhesins—to specific host counterparts. Consequently, adhesins can be used to specifically enrich pathogens. Below, we explore this concept from a variety of viewpoints.

### 1.1. Microbial Adhesion, Colonization and Host Infection

The severity of an infectious disease depends on the level of invasiveness and the extent of host cell and tissue damage caused by the pathogen involved [8]. A varying degree of virulence can be observed across different pathogens and among different strains of a single pathogen [9]. Virulence is a complicated concept in microbial pathogenesis since it depends not only on the infectious agent but also on host cell susceptibility [10]. Adhesion is at the heart of virulence: it plays the initial and decisive role in colonization and subsequent infection (Figure 1) [11,12,13,14]. Bacterial, viral and parasitic pathogens use adhesins to bind to individual host cells and establish interactions with host molecules, thereby initiating colonization. The exact nature of the interaction between pathogen surface molecules and cell receptors defines the cellular or tissue specificity. Such interactions can invoke mechanisms of immune evasion, as adhesion can directly result in the modulation of the host immune response [15]. Finally, not all adhesins have yet been identified, let alone characterized; and, in general, the precise role of adhesins in tissue tropism needs further study. Different bacterial species may target similar or different host ligands using different types of adhesins. At the same time, an individual bacterial species typically harbors multiple adhesion systems with different molecular targets. Understanding and exploiting the molecular basis of pathogen adhesion and the resulting adhesion behavior of whole cells will lead to new formats of diagnostic testing. We propose that cross-disciplinary and translational research is needed to improve our fundamental understanding of adhesion biology, and to translate this knowledge into novel detection strategies based on host–pathogen interaction [16].

### 1.2. Current State of Infectious Disease Diagnostics

The core technologies used by the routine microbiology laboratory are still mostly microscopy- and culture-based. Immunological tests detecting pathogen-specific antigens and antibodies are also routinely used in diagnostics. Furthermore, over the past years, matrix-assisted laser desorption ionization-time of flight mass spectrometry (MALDI-TOF MS) and molecular (nucleic acid-targeting) testing have been introduced successfully in the routine clinical microbiology laboratory (for some recent reviews see [17,18]). 

The main technologies in routine high-throughput laboratories are relatively slow, usually taking at least one overnight incubation. They are limited in terms of sensitivity and specificity, suggesting that there is room for improvement [19,20]. Cultivation methodologies are still internationally accepted as the Gold Standard (Figure 2). Nonetheless these approaches may profit from new, adhesin-based technology to reduce turn-around time and improve test qualities. Actual development of innovative clinical diagnostics requires careful consideration of many parameters, including sensitivity, specificity, cost, shelf-life, robustness, simplicity and user-friendliness (Figure 2 and Figure 3). To validate their quality, new tests must be carefully compared with those from existing diagnostics platforms.

### 1.3. Mandatory Improvement of IVD

Development of better diagnostic tools requires an integrated and interdisciplinary research environment drawing on engineering, biomedical sciences, and product development activities in both academia and the IVD industry [21]. Sharing knowledge, expertise, as well as financial and technical resources, is key to global improvement of the diagnostic field. Innovative start-ups, global players in the clinical diagnostics industry, leading academic institutions, and health care institutions should jointly provide expertise in the domain of medical diagnostics, intellectual property development, marketing, business development and sales. Medical institutions, but also the many (independent) biobanks, should provide access to relevant patient samples (clinical and controls) for verification and validation of new tests [22]. Comprehensive outreach, science communication efforts, and stakeholder engagement are essential to optimize the use of adequate diagnostics. 

An important recent example of the benefits of such an integrated approach was provided during the COVID-19 pandemic. All parties involved generated a huge portfolio of diagnostic tests, immunological and molecular, many of which were rapidly authorized for emergency use (Emergency Use Authorization, EUA) by the US Food and Drug Administration (FDA) [23,24]. The availability of such approved tests allowed for the implementation of high-quality molecular tests during the first wave of the pandemic. Examples are (semi)-quantitative multiplex PCR tests, rapid lateral flow antigen and antibody tests, and most recently the exploitation of next-generation DNA sequencing technologies [23,25]. It has to be noted that this also introduced problems concerning the required capacities of several of these tests, and rapidity of introduction was not always compatible with test quality. Hence, a portion of the immunological tests in particular had to be withdrawn from the market upon accumulation of diagnostic data. 

From a purely technological perspective there are a number of competing approaches that will significantly influence the use of adhesion in clinical microbiological diagnostics. Three of these technologies need a brief assessment. First, PCR and related nucleic acid amplification technologies are sensitive, increasingly cheap, can be deployed widely and essentially detect all microbial species and resistance genes. This technology can be used directly on clinical specimens and it strongly depends on the number of pathogen cells available whether adhesion-based enrichment of such pathogens is required or not. Second, next generation (genome) sequencing (NGS) has become faster and more cost-effective over recent decades, and this is likely to continue. Its performance will soon equal or better that of amplification testing. This suggests that NGS will be routinely used in the microbiology lab. Finally, there is an increase in the rapid availability of high-affinity specific binding reagents other than functional adhesins, such as (monoclonal) antibodies, adhirons and aptamers [26,27]. These have the advantage that they share a basic molecular structure, rendering them suitable for “plug and play” diagnostic applications using the same platforms. If a good diagnostic platform has also been developed, essentially all binding reagents can be applied. Adhesion based assays will have to compete with tests based upon the three concepts mentioned above.

In the following sections, we describe the major interactions between microbial adhesins and host ligands or receptors. We try to define what further structural biology studies are needed, and how these might be useful in the development of novel diagnostic tests.

## 2. Microbial Adhesin–Receptor Pairs

The initial interaction between pathogens and their hosts is defined at the molecular level by the selective interaction of pathogen adhesins with their host receptors. This specific interplay can be exploited in various stages of the classical microbiological diagnostic workflow. To do so, we must extend our understanding of the basic principles of pathogen adhesion so we can apply adhesion assays in the initial capture and enrichment of (complete or parts of) pathogens [28]. All microbial adhesion molecules are surface exposed structures, but their expression may depend on physiological parameters such as environmental temperature, growth stage or availability of nutrients [29,30]. Understanding precisely how microbial adhesins interact with their host receptors poses challenges because the receptor may, for instance, be part of a structurally complex cellular membrane [31]. Site-directed mutagenesis and adhesion assays with whole cells or purified adhesins have shed light on basic aspects of the binding interactions [32]. 

The emergence of SARS-CoV-2 has made it clear that not all adhesins have been identified yet. New ones can be detected using nucleic acid sequencing strategies and searches for new structures that are homologous to known adhesins. Otherwise, proteomic research can be applied to detect cellular surface proteins that provide adhesive characteristics. Random knock-out mutagenesis can also generate cells deficient in adhesion and reverse genetics then allows the functional analysis of the genes and proteins involved. Biophysical technologies can be used to define adhesin structures (see sections below).

### 2.1. Viral Adhesion Processes

Viruses, with relatively small genomes, have a limited repertoire of adhesin structures per individual viral lineage, although an individual adhesin structure is repeated frequently on a single virion. It has to be noted that viruses that are becoming endemic or pandemic exist, even in single hosts, as a species swarm with differing receptor affinities. Recent examples include the SARS-CoV-2 variants such as the B.1.1.7 (α-variant, UK), B.1.351 (β-variant, South Africa), P.1 (γ-variant, Brazilian) and B.617.2 (δ-variant, India) variants of concern [33]. The viral surface is normally quite homogenous allowing for fewer possible receptor specificities [34]. Viral interactions with glycan-based receptors are frequent but typically have affinities in the mM range, while interactions with protein receptors are usually of higher affinity [35,36,37,38,39]. For example, human coronavirus NL63 (HCoV-NL63) uses heparan sulfate proteoglycans (HSPGs) as the initial host receptor. The membrane protein (M) of HCoV-NL63 mediates this attachment to HSPGs and is not spike (S) protein-dependent. It was recently shown that the M protein is also an important player during the early stages of HCoV-NL63 infection, thereby identifying a new adhesin for this virus [40]. Both fungi and viruses exploit a variety of immune modulators to achieve host colonization [41,42]. Recent examples of human receptors relevant for SARS-CoV-2 adhesion are described in Box 1. The emergence of new viruses will undoubtedly lead to the identification of new viral adhesins.

Box 1Adhesion of SARS-CoV-2, the causative agent of COVID-19.The severe acute respiratory syndrome coronavirus-2 (SARS-CoV-2) causes coronavirus disease 2019 (COVID-19). The precise mechanisms of disease are still incompletely understood [43] and replicating virus particles can be observed in a variety of host tissues. Nonetheless, two key host receptors have been identified: angiotensin-converting enzyme 2 (ACE2) [44,45] and liver/lymph node-specific intracellular adhesion molecule-3 grabbing non-integrin (L-SIGN) [46,47]. The crystal structure for ACE2 complexed with the receptor binding domain (RBD) of the SARS-CoV-2 spike protein was solved [48,49]. The binary complex showed clear conservation as compared to similar complexes for the original SARS-CoV-1 virus. This hints at functional conservation of the process of ACE2 binding, but also at the possibility of immunological cross-influences between SARS-CoV-1 and SARS-CoV-2 antibodies. Other forms of structure-based strategies would be the use of the RBD as a subunit vaccine [50] or as a target for inhibition by possible compounds. Neuropilin recognizes a furin cleavage on the SARS-CoV-2 spike protein and is a key target in the development of therapeutics against COVID-19. Recently, X-ray structure-based studies of the neuropilin complexed to the S fragment of the spike protein indicated potentially important design opportunities for therapeutic compounds [13,14]. In addition, virtual drug screening and actual high-throughput screening of compound libraries has been exploited for the key SARS-CoV-2 protease MPRO, which led to the successful identification of potential antiviral drugs [51]. Despite a wide variety of new tests [23], formats based on anti-adhesive strategies have not yet been developed for this priority pathogen.

### 2.2. Modes of Bacterial Adhesion

The nature of the bacterial adhesion molecule varies from distinct organelles such as flagellae or fimbriae to surface exposed, cell wall- or cell membrane-attached proteins, lipids, and sugar (poly- or oligo-saccharide) moieties [52]. Adhesins, especially proteinaceous ones, are often repetitive in primary structure, either by repeating similar domains within a protein chain or by polymerizing subunits into long fibrous structures [53,54]. Most bacterial adhesins tend to bind to host structures that are also often structurally repetitive and ubiquitously distributed, including extracellular matrix (ECM) components such as collagen, fibronectin, or glycoprotein receptors that harbor repeating carbohydrate units.

In certain bacterial adhesins, such as the trimeric autotransporter adhesins (TAAs) [55], the individual binding affinities can be very low (0.1–0.5 M). In this context, binding of pathogens to host cell surfaces is accomplished by avidity, like ‘Velcro’ on a shoe: the three-dimensional arrangement of multiple weak binding sites leads to tight and hence effective binding [56,57,58]. TAAs are can be divided into three domains; a membrane anchored β-barrel domain, a stalk domain and a head domain [55,59,60]. The head domain, once assembled, then adheres to the host ECM via, for example, collagen, vitronectin or fibronectin [58]. Recent work showed that different adhesins bind differently to ECM components and that binding is dramatically influenced by shear forces [56,61,62,63]. In general, improving our understanding of adhesin–receptor interaction requires more detailed insights into their structural aspects [64].

### 2.3. Adhesion Diversity and Evolution

Surface exposed adhesion domains are external moieties and exposed parts of the proteins may be subject to strong environmental selection and possible natural adaptation [65,66]. Hence, the evolution of pathogens is critically linked to the variation in adhesins and their receptor affinities [67], potentially allowing for the colonization of novel hosts. Evolutionary changes in adhesins and ligands can be easily identified by NGS combined with quantitative MS-based proteomics, a combination referred to as proteo-genomics [68,69,70,71]. For instance, conserved peptide sequences (conserved at least within the same species) can be used to perform quantification of species–specific peptides in complex (patient-derived) samples [72,73,74]. The combination of these two methods thus facilitates the correlation of genotypes with adhesion-related phenotypes. There is a continued need for the characterization of additional adhesin–ligand pairs to define conservation of the adhesin or ligand between microbial species. 

## 3. Structural Analysis of Adhesin–Ligand Pairs

There are few structures of bacterial adhesins complexed with their ligands, even though there are many of virus-receptor complexes [35,36]. This may be due to the low-affinity/high-avidity binding of bacterial adhesins, leading to many different complexes and frequent non-specific aggregation. This is problematic because (a) it makes it hard to define the biologically relevant interactions and (b) structural techniques, even cryo-electron microscopy, depend on having a small number (<10) of different conformations and complexes in a single experiment. The modular repetitive structure of some bacterial adhesins and of their host receptors (collagen, laminin, fibronectin, etc.) hampers the determination of their structure and specific interactions by standard methods. Nonetheless, we believe that structural investigations will contribute to the design of better adhesin constructs. These could in return serve as diagnostic tools, as vaccine components, and potentially to develop anti-adhesive drugs.

### Technological and Methodological Developments

NMR spectroscopy, X-ray crystallography, cryo-electron microscopy (cryoEM; see [75,76] for reviews), and mass spectrometry (MS) can be used to characterize the individual adhesin binding domains or their receptors in molecular detail. CryoEM and, to a lesser extent, X-ray crystallography are the best methods for higher-order assemblies. Furthermore, cross-linking MS (XL–MS) and hydrogen-deuterium exchange MS (HDX–MS) are being increasingly used to determine structural constraints between interacting proteins, protein complexes, and their binding interfaces [77,78,79,80,81]. Such constraints can facilitate binding optimization, improvement of ligand design, and thus improved capture for pre-analytical diagnostic steps [82,83,84,85,86,87,88].

Nano-biosensors for miniaturized detection of adhesion, (cryo)EM for visualization of adhesion and identification of molecular partners, X-ray crystallography for the definition of global adhesin structure and more generic tools like NMR for local information on binding partners, and advanced bioinformatics all play essential roles in the further optimization of structure determination and translational applications [89,90,91]. Integrative methods for structure determination of adhesin complexes and for defining their clinical relevance have been shown to be useful [82,83,85,87,92]. For example, structure analysis of *Yersinia enterocolitica* YadA helped to further the understanding of interleukin-1 expression by epithelial cells [93]. The structure of the *Escherichia coli* immunoglobulin binding proteins (Eibs) showed how they are involved in entero–hemorrhagic pathogenicity [94,95]. Another innovative tool that was developed for use in molecular recognition applications was the use of “adhirons” to help stabilize complexes and to gain structural information (e.g., [96]). Adhirons are non-antibody scaffold binding proteins [97]. Well-characterized adhirons display low-nanomolar affinity and high specificity for defined proteins and selectively recognize their target molecules.

## 4. Microbial Adhesion and Future High-Throughput Diagnostic Microbiology Technology

Exploiting the adhesion capacity of pathogens for in vitro diagnostics is a relatively new concept [98,99], but adhesion-based principles can be applied at various stages of the diagnostic process: for specific staining of bacteria, for enhanced species identification, for antimicrobial susceptibility testing, or ultimately maybe even for in vivo therapy (Figure 3). Therefore, prerequisites for further translation into clinical practice are bottom-up research in adhesion from a clinical research perspective, the definition of molecular structure–function relationships, and the design and development of new diagnostic tests and devices [100].

### 4.1. Adhesin-Based Sample Processing in Microbial Diagnostics

Bacterial or viral detection and identification is a complicated multi-step process starting with the collection of clinical samples of diverse origin (blood, sputum, saliva, feces, tears, biopsies, etc.) and consistency (purity, presence of contaminating and possibly test-inhibitory host factors, other microbial species, etc.). Several IVD development projects aim to optimize existing diagnostic tests or to develop novel, specific, and preferably ‘point-of-care’ (PoC) diagnostic tools [101]. We envisage the ability of enriching pathogens from complex samples (e.g., fecal specimens, sputa, or urine samples) to a level where they are free of contaminants and relatively easy to detect by classical tests (Figure 4). This approach is useful in settings with significant sample heterogeneity and contamination, where low numbers of pathogens are present, and where classical clinical microbiology is prone to fail resulting in false-negatives. For instance, *E. coli* O157:H7 has been successfully enriched from contaminated water samples [102]; and it was demonstrated that cell wall binding domains derived from bacteriophage proteins could be used to enrich *Listeria monocytogenes* cells [103]. Pathogen adhesion capacity can be integrated into the pre-analytical sample handling before the actual detection assays to create a unified high-throughput device or protocol.

In many cases the sample needs to be pre-treated in order to prepare it for the actual diagnostic process. Unfortunately, uniform processing methods for samples of diverse origin are rare (for a review see [104] and references therein). Despite the numerous examples listed in Table 2, the development of methods for working with variable sample types and requirements of novel adhesion-based pre-analytical steps for clinical diagnostics are still in their infancy. Further developmental efforts are needed to translate research on adhesion and pathogen capture into diagnostic tests for detection of infections or colonization (e.g., with MDR pathogens).

Progress has been made with certain receptors and bacterial ligands, however. Mannose binding lectin is a host receptor capable of signaling or sensing pathogens exposing mannose at their outer cell surface, and can thus be used to capture a variety of microbial species [105,106]. If mannose binding lectin is attached to a solid surface, mannose-presenting pathogens can be captured on the surface [107]. This approach allows highly sensitive detection of pathogens from a clinical sample at the capturing surface and is applicable in a variety of downstream classical and molecular diagnostic methods. It has been successfully applied in sepsis testing in experimental animals, where an extra-corporal blood-cleansing device was developed to detect pathogens circulating in their blood [108].

A second example concerns protein A (SpA), a surface protein expressed by *Staphylococcus aureus* and other species of coagulase positive staphylococci [128]. SpA has high affinity for the Fc region of IgG antibodies. When immobilized to a solid support, SpA can be used to affinity purify Langerhans cells expressing receptors for the Fc portion of IgG (Fc-IgG), thus generating clean specimens that are well suited for various formats of immune detection [129]. Binding via the Fc part supports the proper presentation of the antigen-binding sites of not only natural but also monoclonal antibodies [108,130]. SpA has proven to be an important biotechnological tool not only in the development of immune tests, but also for the purification and concentration of a variety of human and animal antibodies [131]. However, recent work has shown that SpA does not bind all antibodies uniformly well, an issue that must be kept in mind when developing SpA-mediated protocols [132]. With a similar approach, a 50 amino acid residue termed SAP peptide has been derived from M protein, one of the major virulence factors of *Streptococcus pyogenes.* This can be used to enrich IgA from biological mixtures [133].

### 4.2. Target Enrichment Technology

The development of diagnostic tools requires detailed testing in artificially spiked and “real-life” samples from clinical, environmental, and industrial origins. Institutional or commercially available pathogen strain collections should be used to define the natural variation in the adhesins and the effect of such variation on adhesion efficiency and, hence, test quality [134]. This could even work for an unknown pathogen if the adhesin in question were reasonably well-conserved. The SpA of a *S.* non-*aureus* strain or a new Coronavirus spike protein are significant examples. The question in novel sample enrichment approaches is whether there is a need for test devices that can capture most if not all clinically relevant pathogens or whether sequential testing would be a better option. In any case, new technologies should preferably allow the direct enrichment of pathogens from low-titer samples.

## 5. Clinical–Diagnostic Application of Pathogen Adhesion Tests

High content proteomics [135], electrochemical biosensors [136,137], lanthanide-based fluorescent up-conversion particle assays for detection of adhesin–ligand binding [138], and In Situ Hybridization (ISH) using peptide nucleic acid (PNA) probes and nano-biosensors [139] are only four examples of complex experimental technologies to identify adhesin–receptor interactions, all of which can be translated into new IVD test formats. Such tests can be used in translational research to simplify and accelerate pathogen identification and/or characterization processes (Table 2), which lists tests that are in use summarizing the core competencies and technologies used in these tests.

Currently used physical test platforms range from microfluidic biosensors and nanowires to more classical Raman spectroscopy-, electrochemical- or PCR-based equipment. Furthermore, enrichment platforms frequently utilize nanorods or -wires when straightforward analytical signal detection is required (pathogen present or absent as the final test result). When, after the initial adhesion test, follow-up testing is required in a more preparative manner, magnetic beads are by far the most common approach. The entities to be detected can vary from intact cells, through simple enzymes, to the products of nucleic acid amplification reactions. Of note, nucleic acids rarely play a role in microbial adhesion though biofilms contain relatively large amounts of these molecules and are thought to function as adhesins under biofilm conditions [140]. Still, in diagnosis, nucleic acids are usually employed because they efficiently hybridize to other nucleic acids. These concepts are beyond the scope of the current review.

The most commonly used procedures at the level of detection are coloration, fluorescence measurement, and detection of amplified nucleic acid. The most frequent model organisms used are *E. coli* and *S. aureus*, representatives of Gram-negative and Gram-positive pathogens, respectively. All methods described generate results in 7 min to 3 h, show high sensitivity (to about ten colony-forming units at their most sensitive) and are of great quality; still, their implementation into routine use is sparse. Table 2 and references therein summarize studies that have successfully demonstrated the relevance of adhesion for improved microbiological testing.

### 5.1. Biosensor-Based Pathogen Detection

Biosensors, analytical devices that detect and quantify biomolecules or cells, are composed of three elements: the bioreceptor (allowing binding of the analyte), a transducer (translating the signal into analytical data), and the display set-up [141]. Their advantages are miniaturization, rapidity, mass-production, low cost, high specificity, and automation. This is especially true for electrochemical biosensors where screen-printing of electrodes (SPE) has allowed further miniaturization [142]. Biosensors can be integrated into microfluidic platforms allowing efficient, rapid, portable testing, and permitting reduced volumes of analyte and waste [143]. The basic biosensor consists of a bioreceptor molecule attached to a transducer surface. Upon analyte binding, a recordable change at the transducer surface is measured, usually electrochemical, optical, or mechanical. Different types of bio-receptors are employed for pathogen detection, where antibodies are currently the Gold standard [144]. Other biosensor receptors include, for instance, lectins and bacteriophages or subunits thereof [145]. Most biosensors rely on antibodies or DNA, but the use of adhesins and ECM proteins as receptors for biomolecules or pathogens is increasing. 

There is promising biosensor-mediated research for different microbial pathogens. *E. coli* strain ORN178 can be detected through the binding of type-1 fimbriae to α-D-mannose by attaching the sugar to a nanomechanical cantilever in the biosensor [146]. Upon binding, a change in the resonance frequency of the cantilever is generated. To quantify the adhered bacteria, standard curves displaying the resonance frequencies of the cantilever against bacterial numbers were developed. Biosensors have been developed to study the binding between *E. coli* and ECM proteins in the presence of polysaccharides [147] in a model system, using surface plasmon resonance (SPR) for the inhibition of collagen- and laminin-mediated *E. coli* binding using poly-sulfated polysaccharides. In this work, the binding of pathogenic *E. coli* O157:H7 was also evaluated. SPR and electrochemical impedance spectroscopy (EIS) have been used for rapid detection of *E. coli* through lectin binding using concanavalin A immobilized as a self-assembled monolayer onto a gold electrode surface. These biosensors could be used for screening bacterial load in water samples [148].

*Legionella* collagen-like (Lcl) adhesin binds ECM components and mediates bacterial binding to host cells. Lcl has been used to detect glycosaminoglycans (GAGs) via SPR and EIS onto gold electrodes and gold screen-printed electrodes [149]. The Lcl proteins were immobilized and exposed to different GAGs (Figure 5). Both SPR and EIS could detect high-affinity binding of GAGs. This shows that both techniques can be used for the diagnosis of *L. pneumophila* lung infection. In addition, haemagglutinin (HA), a homo-trimeric glycoprotein expressed on the surface of the influenza virus, shows high affinity towards sialic acid-terminated trisaccharides of epidermal cell membranes. Researchers developed both a mechanical and an electrochemical biosensor for the detection of the human influenza virus based on this interaction [150]. Binding of 2,6-sialyllactose to HA could be detected in a label-free manner via impedance using a Quartz Crystal Microbalance (QCM).

### 5.2. Next Generation Test Formats—Bioactive Surfaces and Materials

There is still an urgent need both for improved insight into how to capture and enrich pathogens from complex clinical samples for downstream analytical diagnostics, and for the design and study of anti-adhesive compounds (to help prevent non-specific binding) [151]. For instance, do tests exploiting multiple adhesins for target enrichment perform better than assays using a single adhesin or not? What volume of clinical sample is sufficient or needed for adequate diagnostics? Complex clinical specimens (sputa, fecal material, blood) will be much more difficult to work with than, for instance, simple ones such as infected urine with relatively high numbers of pathogens present per unit of volume. In many cases it is also not yet known what the duration of the capture step should be, nor what the costs of an assay are going to be. For IVD manufacturers, knowing the answers to such questions is essential in the decision-making process preceding the development of an adhesion assay.

Current diagnostics for bacterial and viral pathogens are typically unspecific and include cultivations of uncertain sensitivity, as pathogen concentrations as low as one viable count per mL can be indicative of infection [152]. Enrichment directly from a patient sample can speed up pathogen diagnostics by many hours and increase sensitivity, but may render downstream analysis of testing for, e.g., antibiotic resistance somewhat complex [153]. Specific ligands (e.g., peptides, peptide nucleic acids, glycans, and aptamers) that bind to ECM, membrane components, or capsids of (a) given pathogen(s) can be immobilized onto the surface of materials used for fabrication of diagnostic devices, such as peptide or nucleic acid arrays, microbeads, membranes, and even electrodes using a paper format. Such bioactive surfaces provide sufficiently high and well-controlled binding capacities for ligands, intact cells, and cellular extracts. The surface characteristics prevent denaturation of the immobilized ligands, allow for convenient and efficient immobilization techniques, and are preferably reversible to allow regeneration. Such surfaces should prevent non-specific interactions, i.e., be anti-adhesive or anti-fouling or even prevent infection, which is particularly relevant for intra-corporeal devices [154]. A recurring problem is the significant fraction of bacteria that is accidentally lost by non-specific binding in miniaturized devices and microfluidics due to their high surface area to volume ratios. Thus, antifouling properties are of greater importance [155]. Furthermore, fouling by non-specific biomolecules can also hinder sensitivity and selectivity and result in false-positive or -negative readings [156]. Improved materials can be obtained by physico–chemical modification of the surface [157], grafting [158,159], coating [160], surface topography modification [161] and surfactant adsorption [162]. At present, the development of novel multi-functional materials to capture pathogens from biological samples combining grafting with plasma or UV treatments will help integrate the pre-enrichment methods into full diagnostics workflows [163,164,165]. Overall, this approach can lead to fast, specific, and efficient pathogen trapping strategies, reduced sample processing times, and better sensitivity for downstream detection techniques. The inverse process, anti-adhesion, can be used to develop “clean” materials (see Box 2).

Box 2Anti-Adhesion.The development of anti-adhesive materials with minimal fouling, based on ‘grafting-from’ approaches, may be useful to inhibit adhesion. The development of ‘anti-ligands’ to inhibit adhesion provides additional therapeutic approaches. Pilicides, which inhibit the first steps in biofilm formation for *E. coli*, constitute a special category of such anti-ligands [166,167], and these are being considered as alternative therapeutic approaches in, for instance, bacterial urinary tract infections [168]. The target of pilicides is the biofilm, but some also show surprising anti-pilin activity, thereby tackling infections from two fundamentally different angles [169]. Their diagnostic value is limited at this stage although detection of pilicide activity could indicate early stage biofilm formation. Altogether this could provide important tools for the prevention of (nosocomial) infections in general. Simultaneous detection of multiple pathogens may thus be a distinct possibility. This will facilitate new formats for syndrome testing, where all possible pathogens involved in a certain type of infection can be detected at the same time and ruled in or out simultaneously (e.g., gastro–intestinal or respiratory infections) [170,171,172]. This has clear benefits both in terms of cost, morbidity, and even mortality, as the faster the correct pathogen(s) are identified, the faster the correct treatment can be given.

## 6. Conclusions

The IVD workflows show the overall state of readiness for innovation in clinical microbiology (Figure 3 and Figure 4). Expansion of the existing portfolio of IVD tests is a must, and innovative adhesion-based assays have been proposed. Some of these are already accepted for routine diagnostic use, but there are still many procedures in development that require additional validation, verification and, in the end, user acceptance as valuable IVD tools. Acceptance of such tests will improve overall public health status by helping control the spread of pathogens and allowing for personalized treatment. We believe that the application of integrative approaches, including bioinformatics, quantitative and structural proteomics and structure modeling, will improve the understanding of the pathogenesis of infectious diseases in general [173]. Understanding the sequential and structural determinants of adhesion will further drive the translational aspects, leading to the design of novel tests. New technologies, as described here, will play a key role in facilitating this essential phase of test design [174,175,176,177]. Cost effective scale-up and application of adhesins to commercially relevant sensor-activated testing systems needs to be implemented in the developmental cycles exploited by commercial companies.

COVID-19 has, at last, made the general public aware of the global impact of infectious diseases and the need for and relevance of rapid diagnostics [178]. We must seize the moment: this current appreciation of the value of infectious disease testing should be used to push for affordable, high-quality, rapid diagnostic tests for all infectious diseases to be available worldwide. We believe that adhesion research will contribute to this, since it defines fundamental new processes that allow identification and enrichment of new binding partner molecules. Such molecules can then be implemented in IVD using the continuously expanding experimental toolbox to allow sensitive detection of molecular binding events. The combination of accelerated detection and identification of microbial adhesins and the technological capabilities defines a prosperous future for this field of in vitro diagnostics.

## Figures and Tables

**Figure 1 diagnostics-11-01259-f001:**
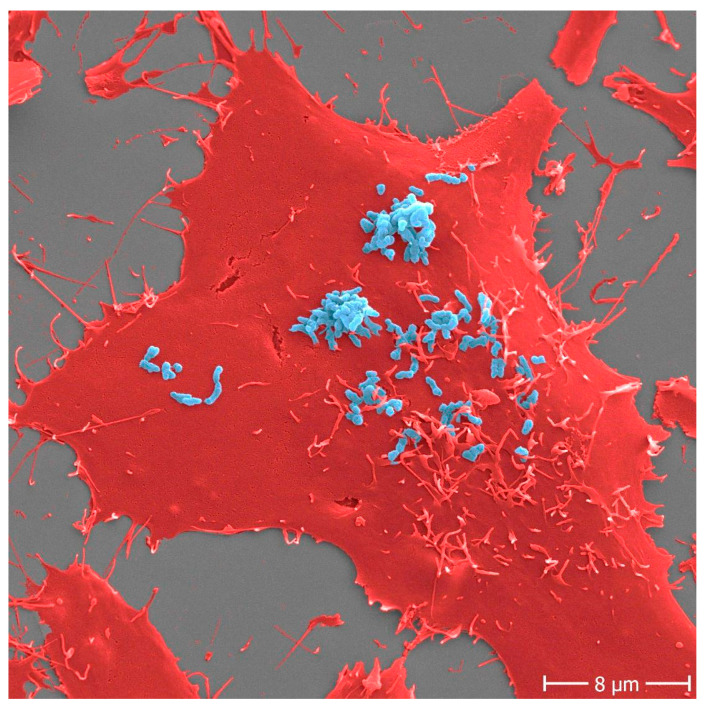
Adhesion of *Bartonella henselae* to human cells. *B. henselae* (strain Marseille) bacteria (light blue) in an early stage infection process (30 min) to human HeLa-229 cells (red). Adhesion to host cells is mediated by specific interactions between *B. henselae* surface proteins and components of the host extracellular matrix including molecules such as fibronectin or collagen. Scale bar: 8 μm.

**Figure 2 diagnostics-11-01259-f002:**
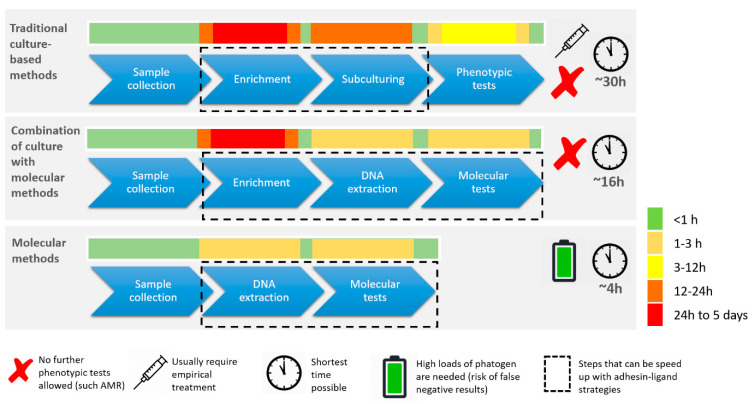
Comparison of culture-based versus molecular diagnostics in routine clinical microbiology. Innovative clinical diagnostics with inclusion of molecular methodology requires careful consideration of many important test parameters among which sensitivity, specificity, cost, shelf life, robustness, simplicity, and user-friendliness. Although molecular techniques are faster, there is frequently a mandatory need for cultivation throughout the diagnostic process. Viable cells are often required for storage, downstream AST or simply for reproducibility testing at a later stage.

**Figure 3 diagnostics-11-01259-f003:**
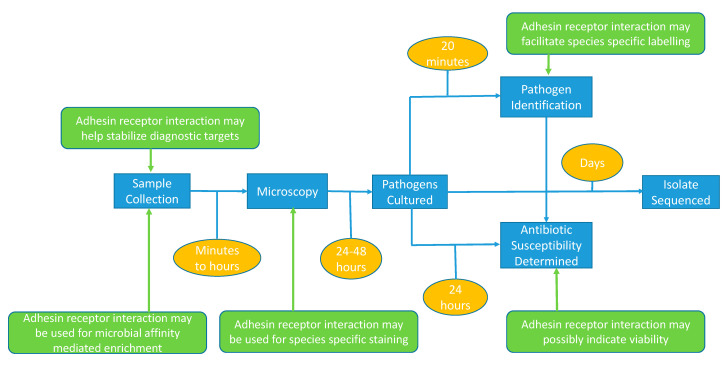
Diagnostic workflow and technologies (blue boxes) used for the routine detection of pathogens. Current timing is indicated in orange boxes whereas the possible impact of adhesion-based assays is indicated in green.

**Figure 4 diagnostics-11-01259-f004:**
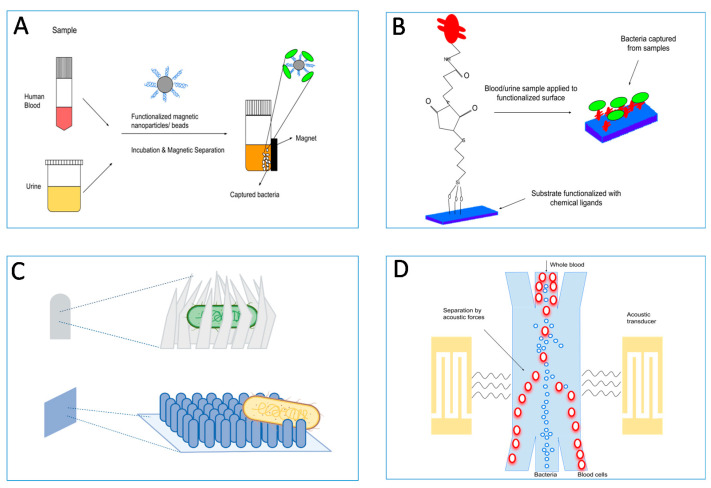
Adhesin–ligand interactions can be used to enrich pathogens from complex mixtures (e.g., fecal specimens, sputa, or urine samples) to a level where they would be clean and relatively easily detectable by classical tests. (**A**) Most common pre-enrichment methods make use of functionalized magnetic particles such as nanoparticles or beads, which bind to pathogens present in biological samples and are afterwards separated magnetically. (**B**) Surfaces functionalized with chemical cross-linkers and affinity ligands are used to directly capture bacteria with high specificity from biological samples. (**C**) Various types of nano-topographies such as prickly or nano-patterned surfaces, or nano-claws are used to capture bacteria. They are used alone or in combination with capture ligands. (**D**) Separation of bacteria from blood cells using surface acoustic waves in a microfluidic device. Other separation techniques such as viscoelastic separation are also used in microfluidic devices.

**Figure 5 diagnostics-11-01259-f005:**
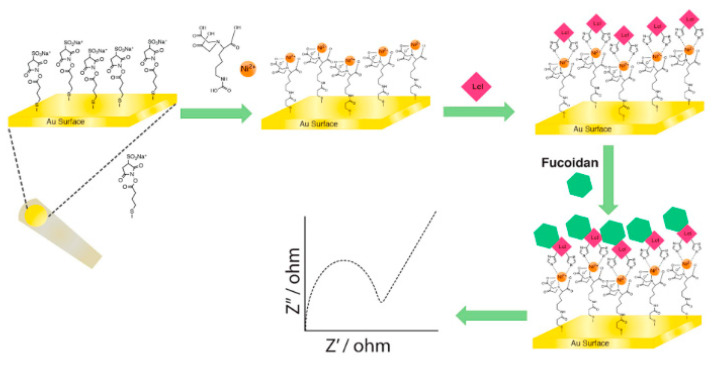
Illustrative representation of Lcl immobilization and fucoidan detection using EIS. A gold electrode was chemically modified after which nickel was electrostatically bound to the surface. This facilitated the binding of *Legionella* proteins, which were detected by electrochemical impedance spectroscopy (EIS). Reproduced with permission from [149].

**Table 1 diagnostics-11-01259-t001:** WHO priority bacterial pathogens and their clinically relevant antibiotic resistance phenotypes which render treatment problems (adapted from www.who.int (accessed on 21 June 2021) and [7]). ESBL: extended spectrum beta-lactamase.

**Priority 1: Critical**	
*Acinetobacter baumannii*	carbapenem-resistant
*Pseudomonas aeruginosa*	carbapenem-resistant
Enterobacteriales	carbapenem-resistant, ESBL-producing
**Priority 2: High**	
*Enterococcus faecium*	vancomycin-resistant
*Staphylococcus aureus*	methicillin-resistant, vancomycin-intermediate and resistant
*Helicobacter pylori*	clarithromycin-resistant
*Campylobacter* spp.	fluoroquinolone-resistant
*Salmonellae*	fluoroquinolone-resistant
*Neisseria gonorrhoeae*	cephalosporin-resistant, fluoroquinolone-resistant
**Priority 3: Medium**	
*Streptococcus pneumoniae*	penicillin-non-susceptible
*Haemophilus influenzae*	ampicillin-resistant
*Shigella* spp.	fluoroquinolone-resistant

**Table 2 diagnostics-11-01259-t002:** Review of pre-analytical target enrichment methods using adhesion receptor interactions to detect infection. Note that samples mostly consisted of artificially spiked materials. Hence, most of the tests target bacteria. The test costs could not be compared, as data were frequently missing. (LPS: Lipo-Poly-Saccharide; LTA: Lipo-Teichoic Acid).

Device	Enrichment Technique	Target	Detection Method	Limit of Detection	Time	Reference
3D printed microfluidic biosensor	Aptamer coated magnetic beads with magnetic separation	*Plasmodium falciparum* lactate dehydrogenase (PfLDH) enzyme	Colorimetric	Parasitemia < 0.01%	180 min	[109]
Enzyme-linked LTF assay (ELLTA)	Long tail fibers (S16 LTF) of bacteriophages immobilized onto paramagnetic beads	*Salmonella typhimurium*	Colorimetric	10^2^ cfu/mL	2 h	[110]
Assay	Magnetic beads coated with the engineered chimeric human opsonin protein, Fc-mannose-binding lectin (FcMBL)	Articular fluid samples and synovial tissue samples from patients with *S. aureus* infections	RT-PCR analysis and MALDI-TOF	76% ± 5.7% capture efficiency	-	[111]
Assay	Iron oxide magnetic nanoparticles functionalized with bacterial species-identifiable aptamers	*S. aureus* and *E. coli*	Fluorescence microscopy	10 CFU	1.5 h	[112]
Microfluidic platform	Induced advectivespiral flows of super-paramagnetic nanoparticles coated with mannose-binding lectin and magnetic separation	*E. coli* spiked into undiluted rat whole blood	None	91.68% ± 2.18% capture efficiency	-	[113]
3D Nano-biointerface platform	Zinc oxide nanorod array 3D nano–bio surface functionalized with lectin Concanavalin A	*E. coli*	Fluorescence microscopy imaging	0.9 × 102 CFU/mL	-	[114]
Nanowire arrays	Functionalized 3D nanowire substrate	*S. aureus*	Fluorescence microscopy	10 CFU/mL	30 min	[115]
Nanowire arrays	Bendable polycrystalline nanowires pre-grafted on 3D carbon foam	Human blood spiked with *Salmonella spp*	Fluorescence microscopy	~97% capture efficiency	-	[116]
Impedanceelectrode sensor	Antibacterial prickly Zn-CuO nanoparticles with burr-like nanostructures	Rat blood spiked with *E. coli*	Impedance-based electrode sensor	10 CFU/mL	20 min	[117]
Surface-Enhanced Raman Scattering Multi-Multifunction Chip	4-mercaptophenylboronic acid	Humanblood spiked with *E.coli*, *S. aureus*	Surface-Enhanced Raman Scattering	1.0 × 10^2^ cells m/L	-	[118]
Photoelectrochemical platform	4-mercaptophenylboronic acid	*E. coli*	Photoelectrode	46 CFU/mL	30 min	[119]
Microfluidic platform	Magainin 1 peptide	urine spiked with *Salmonella spp*; *Brucella spp*	Recombinase polymerase amplification (RPA) sensor	5 CFU/mL urine for *Salmonella*; 10 CFU/mL for *Brucella*	60 min	[120]
Microfluidic chip	Bulk acoustophoresis	diluted whole blood spiked with *Pseudomonas putida*	Microscopy	-	12.5 min	[121]
Microfluidic chip	Bulk acoustophoresis	*Pseudomonas aeruginosa*, *S. aureus, E. coli*	Luminescent bacterio-phage assay	45% to 60% capture efficiency	-	[122]
Microfluidic capillaric circuit	Antibody-functionalized microbeads	synthetic urine spiked with *E. coli*	Fluorescence microscopy	1.2 × 10^2^ CFU/mL	7 min	[123]
Microfluidic chip	Pillar-assisted self-assembly microparticles Nano- filter for	*E. coli* from samples	Fluorescence microscopy	capture efficiency of 93%	-	[124]
Reusable supramolecular platform	Multilayered film and β-cyclodextrin (β-CD) derivatives modified with mannose	Type I fimbriae *E. coli* and lectin proteins	Fluorescence microscopy	Capture efficiency of 93%	-	[125]
Photonic PCR on a chip	Gravity-driven cell enrichment	*E. coli*	Photonic PCR on a chip	10^3^ CFU/mL	10 min	[126]
Enzyme-linked lectin sorbent assay (ELLecSA)	Fc-mannose-binding lectin	Bacteria, fungi, virus, parasites. LPS, LTA from Gram-negative and Gram-positive bacteria, as well as lipo-arabino-mannan (LAM) and phosphatidyl-inositol mannoside from *M. tuberculosis*	Scanning electron microscopy	-	<1 h	[114]
Fluorometric assay	Two distinct terminal phosphate-labeled LPS specific aptamers attached onto Zr-MOFs to fabricate the magnetic core-shell for magnetic separation	*Acinetobacter baumannii* in blood samples	Fluorescent signal amplification by fluorescence probes	10 cfu/mL	~2.5 h	[127]

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
