# Peer review of "Host-Pathogen Adhesion as the Basis of Innovative Diagnostics for Emerging Pathogens"

_diagnostics, 2021, doi:10.3390/diagnostics11071259_

Round 1
Reviewer 1 Report
This review was well written and covered significant summary of current research on diagnosis of pathogens based on adhesins. I do not see any problems with the manuscript.
Author Response
We thank the reviewer for his or her positive assessment. Nothing was change din the text since no changes were asked for.
Reviewer 2 Report
This review provides an inventory of methods based on adhesion factors as well as pathogen attachment mechanisms. The traditional in vitro diagnostic methods often suffer challenges for enrichment and sensitivity issues. So the method using the adhesion factors can be also of relevance to currently emerging pathogens, including SARS-CoV-2, the causative agent of COVID-19. The review was organized well and presented in a clean manner. I have a few general comments:
- I am not an English speaker. But when I read through the manuscript, I felt it can be improved with the help of a native English speaker.
- It will be nice to provide a Figure for 2.2.
- Some of the clinical diagnostic output aspects based on adhesion could be more detailed. My overall impression is that authors spent a lot of efforts in mechanisms rather than diagnostics application.
- Any molecular amplification method is related to adhesion?
Author Response
We have reacted point by point to the comments raised by this reviewer below (in italics). We thank the reviewer for the constructive assessment.
This review provides an inventory of methods based on adhesion factors as well as pathogen attachment mechanisms. The traditional in vitro diagnostic methods often suffer challenges for enrichment and sensitivity issues. So the method using the adhesion factors can be also of relevance to currently emerging pathogens, including SARS-CoV-2, the causative agent of COVID-19. The review was organized well and presented in a clean manner. I have a few general comments:
- I am not an English speaker. But when I read through the manuscript, I felt it can be improved with the help of a native English speaker. We have implmented another round of serious editing and the result of that can be seen in the numerous changes in the annotated V2 version of our paper that is being submitted herewith.
- It will be nice to provide a Figure for 2.2. We decided not to include an illustration. First, we have already quite a few in the paper and, second, there is already a lot in the international literature. We did however add reference to a paper that does show such illustrations (Stones and Krachler, 2015) and we hope this is OK with the reviewer.
- Some of the clinical diagnostic output aspects based on adhesion could be more detailed. My overall impression is that authors spent a lot of efforts in mechanisms rather than diagnostics application. We think that htis question of the reviewer relates to the text mostly. We agree that this was a bit short but the actual tests are described in the Table 2. And there is a lot of information on test formats there. So we decided to put mor emphasis on Table 2 by expanding the reference to it in our text. So we did not add more test description but tried to make clear that this information is summarized in Table 2.
- Any molecular amplification method is related to adhesion? We have now more explicitly stated in the text that nucleic acids are usually not considered as adhesins, except for maybe during the biofilm lifestyle. We als state that, of course, nucleic acid testing (incliuding amplifcation) can be used for a read out system for adhesion assays.